# Seed Pelleting with Gum Arabic-Encapsulated Biocontrol Bacteria for Effective Control of Clubroot Disease in Pak Choi

**DOI:** 10.3390/plants12213702

**Published:** 2023-10-27

**Authors:** Rizwangul Abdukerim, Sheng Xiang, Yanxia Shi, Xuewen Xie, Lei Li, Ali Chai, Baoju Li, Tengfei Fan

**Affiliations:** State Key Laboratory of Vegetable Biobreeding, Institute of Vegetables and Flowers, Chinese Academy of Agricultural Sciences, Beijing 100081, China; reziwan52@163.com (R.A.); xiangsheng@caas.cn (S.X.); shiyanxia@caas.cn (Y.S.); xiexuewen@caas.cn (X.X.); lilei01@caas.cn (L.L.); chaiali@caas.cn (A.C.)

**Keywords:** *paenibacillus polymyxa* ZF129, gum arabic, clubroot, seed pelleting, microencapsulation

## Abstract

Clubroot is one of the most serious soil-borne diseases on crucifer crops worldwide. Seed treatment with biocontrol agents is an effective and eco-friendly way to control clubroot disease. However, there is a big challenge to inoculating the seed with bacterial cells through seed pelleting due to the harsh environment on the seed surface or in the rhizosphere. In this study, a method for microbial seed pelleting was developed to protect pak choi seedlings against clubroot disease. Typically, a biocontrol bacterium, *Paenibacillus polymyxa* ZF129, was encapsulated by the spray-drying method with gum arabic as wall material, and then pak choi seeds were pelleted with the microencapsulated *Paenibacillus polymyxa* ZF129 (ZF129m). The morphology, storage stability, and release behavior of ZF129 microcapsules were evaluated. Compared with the naked *Paenibacillus polymyxa* ZF129 cells, encapsulated ZF129 cells showed higher viability during ambient storage on pak choi seeds. Moreover, ZF129m-pelleted seeds showed higher control efficacy (71.23%) against clubroot disease than that of nonencapsulated ZF129-pelleted seeds (61.64%) in pak choi. Seed pelleting with microencapsulated biocontrol *Paenibacillus polymyxa* ZF129 proved to be an effective and eco-friendly strategy for the control of clubroot disease in pak choi.

## 1. Introduction

Clubroot caused by the soil-borne obligate parasite *Plasmodiophora brassicae* Woronin is one of the most devastating diseases on Cruciferae crops. The roots of the infected plants by *P. brassicae* will grow in the form of large galls, which can disrupt water and nutrient uptake and eventually result in the wilting and stunting of plants [1]. Clubroot causes 30% of yield losses annually worldwide, thus affecting more than 70 countries [2]. In addition, this disease is difficult to manage because of its great variation in pathogen virulence and the long survival (up to 20 years) of resting spores of *P. brassicae* in the soil [3]. Therefore, several strategies have been developed to control clubroot, including the utilization of resistant cultivars, crop rotations, liming of soils, chemical fungicide treatments, and biocontrol agent treatments. Typically, the application of biocontrol microbes is more economical and ecologically friendly than chemical treatments and more time-saving than breeding resistant cultivars.

Naturally occurring microorganisms have the potential to reduce clubroot [4,5,6]. Some of these organisms produce anti-microbial metabolites against *P. brassicae* [7,8], while others colonize roots [9,10] and induce resistance to the disease [11]. The microbial control of clubroot is attractive because certain soil microbes can colonize the root or the rhizosphere and thus potentially provide durable protection. In the previous study, *P. polymyxa* ZF129 was proven to be effective in the control of soil-borne diseases [12]. However, inoculation of *P. polymyxa* ZF129 into the environment may result in a poor survival rate because of their susceptibility to various environmental factors. Thus, encapsulation of bacteria using biocompatible wall materials is an effective strategy to protect sensitive bacteria from unfavorable environments. In addition, microcapsules could be designed to deliver core materials in a controlled manner by bursting, diffusion, and dissolution, depending on the properties of the wall materials. Among the microencapsulation techniques, the spray-drying method is the most widely used in biocontrol industries because of its low operation cost and high production rate. However, the activity of the biocontrol bacteria could be negatively affected by various parameters, such as the high temperatures.

Seed inoculation has been considered a precise and cost-effective method to deliver microbial inoculants [13,14], with the potential for large-scale application. Seed coating has been proposed as a promising tool to inoculate different crop seeds because it can utilize minor amounts of inocula in a precise application [15,16,17,18,19,20,21]. Seed coating is a technique in which an active ingredient (e.g., microbial inoculant) is applied to the surface of the seed with the help of a binder and, in some cases, a filler that can act as a carrier. The main types of seed coatings include seed dressing, film coating, and pelleting, which can be chosen differently based on the purpose of application and the type of seed or selected microbes [22].

Pak choi (*Brassica rapa* L. ssp. *chinensis* L.) is one of the most popular vegetables in China, with annual consumption accounting for 30–40% of the total annual vegetable consumption in eastern China. However, in recent years, clubroot has caused increasing damage to pak choi, thus affecting its production. Therefore, this research aimed to propose a biocontrol method through pak choi seed pelleting with an encapsulated *P. polymyxa* ZF129. The physical and biological properties of pelleted pak choi seeds were systematically investigated, including the degree of uniformity, germination rate, storage stability, and control efficacy of clubroot. The results of this study would provide a theoretical basis and technical support for the application of biological seed treatment to vegetable crops.

## 2. Materials and Methods

### 2.1. Material and Bacterial Strain

*P. polymyxa* ZF129 strain (GenBank CP040829.1) was provided by the Institute of Vegetables and Flowers, Chinese Academy of Agricultural Sciences (Beijing, China). Herein, 0.3% *w*/*w* of aqueous-carboxymethyl cellulose (CMC) (Sigma Aldrich, St. Louis, MO, USA) was prepared as the binding agent. Gum arabic powders were purchased from (Sinopharm, Beijing, China). Pak choi (*Brassica rapa* L. ssp. *chinensis* L.) seeds were obtained from the Institute of Vegetables and Flowers, Chinese Academy of Agricultural Sciences (Beijing, China).

### 2.2. Bacterial Strains and Culture Conditions

Bacterial stocks of *P. polymyxa* ZF129 used for the inoculation were stored at 20 °C in Lysogeny Broth (LB) with 20% (*v*/*v*) glycerol. The *P. polymyxa* ZF129 strain was cultured in the LB at 30 °C with a shaking rate of 180 rpm in the dark for 48 h. Cell concentrations were determined by following the absorbance at 660 nm.

### 2.3. Preparation of P. polymyxa ZF129 Microcapsules by Spray-Drying

The 48-h *P. polymyxa* ZF129 cell cultures with a cell concentration of 10^8^ CFU/mL were mixed with 50% (*w*/*v*) of gum arabic solution in a 1:1 ratio. Then, the mixture was spray-dried in a laboratory spray dryer with an airflow and feed flow of 500 L/h and 6 mL/min, respectively. Subsequently, the inlet temperature was set independently at 120 °C, 180 °C, and 200 °C. Afterward, the sample mixture was agitated using a magnetic stirrer at room temperature during the whole process. Finally, the dried microcapsule powder was collected in a Schott glass bottle and stored at 4 °C before further analysis.

### 2.4. Preparation of P. polymyxa ZF129 Pelleted Seed (ZF129 Seed) and P. polymyxa ZF129 Microcapsule Pelleted Seeds (ZF129m Seed)

Four types of filler materials (peat, talcum, bentonite, and diatomite) were chosen as inert pelleting fillers, and their combination was designed by DesignExpert v11 (Table 1). Herein, 0.3% (*w*/*v*) of aqueous carboxymethyl cellulose (CMC) solution was prepared as a binding agent. For *P. polymyxa* ZF129 microcapsule pelleted seeds (ZF129m seed), 100 g of pak choi seeds, 300 g of pelleting powder containing 10 g of ZF129 microcapsules (10^8^ CFU/g), and 290 g of inert filler materials were placed in the rotary coater, and rotation speed was set at 90 rpm. During the pelleting, pak choi seeds were wetted with the binder, which was dropped through an atomizer. Meanwhile, for the *P. polymyxa* ZF129 pelleted seed (ZF129 seed), a volume of the *P. polymyxa* ZF129 culture containing the same number of cells with the abovementioned microcapsules was concentrated via centrifugation to pellet the cells. In the binding agent, the supernatant and cells were respectively decanted and resuspended. Then, cells were embedded in the filler matrix through wetting. The pelleting seeds were dried at 35 °C for at least 0.5 h before being used. Stiffness testing was conducted on 10 pelleted seeds per treatment using a tablet hardness meter (Vankel^®^, Clark, NJ, USA). For each recipe, a total of 100 pelleted seeds were planted in multi-cavity trays that contained 100 g of sterilized commercial substrate soil (purchased from Weiwei Horticulture Company, Beijing, China). The germination rate of seeds was recorded daily until a constant germination rate was achieved, with untreated seeds serving as the germination control.

### 2.5. Field Emission Scanning Electron Microscopy (FESEM)

The morphology of *P. polymyxa* ZF129 microcapsules was characterized by field emission scanning electron microscopy (JEOL, JSM-7600F, Tokyo, Japan). The samples were spread onto conductive carbon tape and then stuck on a SEM stub. Furthermore, samples were sputter-coated with gold using a JFC-1600 Auto Fine Coater (JEOL, Tokyo, Japan) to a thickness of 10 nm. Afterward, samples were imaged at different magnifications with an acceleration voltage of 3.00 kV.

### 2.6. Dynamic Laser Scattering

The particle size distribution of *P. polymyxa* ZF129 microcapsules was determined using a dynamic laser scattering (DLS) instrument (Zetasizer Nano ZS90, Malvern Panalytical, Malvern, UK). Ethanol was used as a dispersant for microcapsules [23]. The size distribution and the mean size of microparticles (volume and number distributions) were determined for three 30-s runs. Each sample was measured in triplicate.

### 2.7. Determination of Moisture Content

Weighing dishes were initially dried in an oven at 105 °C to a constant weight and then cooled in a desiccator containing silica gel. Then, the weight of the empty dish was recorded (a), and approximately 3 g of *P. polymyxa* ZF129 microcapsule powder was added, and the dish was reweighed (b). The loaded dish was placed in the oven at 105 °C for 24 h, then cooled to room temperature in a desiccator and reweighed (c). The heating and cooling processes were repeated until the weight (c) was constant [24].
Water content = (b − c)/(c − a) × 100% 

a = weight of an empty dish;b = weights of dish and wet powder;c = weights of dish and dried powder.

### 2.8. Encapsulation Efficiency of P. polymyxa ZF129 Microcapsules

Briefly, 1 g of spray-dried *P. polymyxa* ZF129 microcapsules was introduced in 9 mL of deionized distilled water. Then, the sample was serially diluted in distilled water and plated on the LB medium. After 48 h of incubation at 37 °C, cell counts were determined, the survival of encapsulated bacteria was evaluated, and the number of colony-forming units per gram (CFU/g) was measured.
Encapsulation efficiency (EE) = (Log_10_ N/Log_10_ N_0_) × 100 
where N is the number of entrapped viable cells and N_0_ displays the free viable cells before encapsulation.

### 2.9. Storage Stability of P. polymyxa ZF129 Microcapsule Powder and P. polymyxa ZF129 Pelleted Seeds

For the *P. polymyxa* ZF129 microcapsule powders, samples were placed in tightly sealed sterile bottles and stored for 30 days at 4 °C and 25 °C. The viability of cells at different storage times (0, 7, 14, 21, and 30 days) was determined using the colony counting method.

In addition, for the *P. polymyxa* ZF129-pelleted seeds, including ZF129 seeds and ZF129m seeds, seed samples were placed in tightly sealed sterile bottles and stored for 60 days at 25 °C. A total of 10 seeds were placed in 10 mL of sterilized water and then shaken to allow bacteria to release from the seed pelleting matrix. Then, the enumeration of *P. polymyxa* ZF129 within the suspension was conducted using the colony counting method.

### 2.10. Release Behavior of P. polymyxa ZF129 Cells from Pelleted Seeds

The release behavior of the *P. polymyxa* ZF129 cell from pelleted seeds was studied based on the protocol described by Cortés-Rojas et al. [25]. with little modification. First, a soil extract-based release medium was prepared as follows: 100 g of the native soil was suspended in a 500 mL sterile PBS buffer at pH 7.4 and then incubated at 40 °C with agitation at 200 rpm for 1 h. Then, the extract was filtered to separate coarse particles and on filters with a 0.2-μM pore size to further remove other fine solids [26]. Second, five pelleted seeds were placed in a 5 mL soil extract solution inside a sterile 50 mL polypropylene tube. Subsequently, the release medium was taken out at different preset time points (1 h, 5 h, 24 h, 48 h, and 72 h) to determine the release rate of ZF129 from each batch of pelleted seeds. The evaluation was carried out in triplicate.

### 2.11. Efficiency of P. polymyxa ZF129 Microcapsule Pellet Seed against Clubroot on Pak Choi

*P. polymyxa* ZF129 microcapsule pelleted seeds (ZF129m seeds), ZF129 suspension pelleted seeds (ZF129 seeds), and non-treated seeds were sown in trays (540 mm × 280 mm) containing 4000 g of *Plasmodiophora brassicae* Woronin (*P. brassicae*)-infected soil (10^8^ spores/g soil). The *P. brassicae* inoculum used in this study belongs to pathotype 4. The pak choi plants were cultured in the greenhouse with the following conditions: 25 °C, 16-h photoperiod, and 70% RH. After 60 days, all plants were harvested, and the clubroot severity was rated using a slightly improved grading standard [27], which included 0–3 scales: 0 = normal root growth without galling, 1 = galls on main roots or a few small galls formed on <1/3 lateral roots, 2 = galling on the main root or 1/3–2/3 lateral roots, and 3 = larger galls formed on the 2/3 of main and lateral roots. The disease severity index (DSI) and control efficacy were calculated using the following equations [28].
DSI (%) = [(Σ [rating class × number of plants in the rating class])/(total number of plants in the treatment run × highest rating scale)] × 100.
Control efficacy(%) = (DSI of infected control) − (DSI of treatment)/(DSI of infected control) ×100%

### 2.12. Statistical Analysis

Data were subjected to analyses of variance (ANOVA) using SPSS 13.0 software (SPSS, Inc., Chicago, IL, USA). Mean comparisons were conducted using a least significant difference (LSD) test at a 0.05 probability level.

## 3. Results and Discussion

### 3.1. Morphology of Microcapsules

The morphology of *P. polymyxa* ZF129 microcapsules prepared at different inlet air temperatures is shown in Figure 1. As shown, *P. polymyxa* ZF129 microcapsules prepared at the inlet air temperature of 120 °C showed a spherical shape with small wrinkles on the surface. As the inlet temperature increased from 180 °C to 200 °C, the surface of the microcapsule appeared smooth; however, the shape became irregular, and more concavities were observed. These concavities were also reported in other studies using gum arabic as a wall material. Thus, it can be explained that concavities were caused by the rapid evaporation of the atomized liquid drops during spray drying [29]. Notably, no fractures or debris were observed on all three samples, thus indicating the ability of the wall material to withstand mechanical forces associated with expansion and ballooning during spray drying [30].

### 3.2. Filler Formula Screening for Seed Pelleting

The physical properties, such as single seed rate, abscission rate, and disintegration rate, were tested for pelleted pak choi seeds to screen the filler formula for seed pelleting, which was prepared based on the 11 filler formulas (Table 2). Among these, formula nine showed the best performance in terms of a high single seed rate (100%) and disintegration rate (100%) as well as a low abscission rate (5%). In addition, the effect of the filler formula on the germination rate of pak choi seeds was also investigated. The result showed that the mean germination rate was significantly affected by the filler formula, ranging from 4.80% to 96.90%, whereas formula nine still performed the best (96.90%). Therefore, formula nine was selected as the best filler formula for pak choi seed pelleting.

### 3.3. Encapsulation of P. polymyxa ZF129 by the Spray Drying Method

*P. polymyxa* ZF129 was encapsulated by the spray drying method using gum arabic as the wall material under different inlet air temperatures (120 °C, 180 °C, and 200 °C). The counts of viable cells before and after spray-drying treatment are shown in Table 3. The inlet air temperature increased from 120 °C to 200 °C, encapsulation efficiency decreased from 97.00% to 96.86%, 96.17%, and bacterial loading amount also decreased from 8.09 ± 0.05 log CFU/g to 7.04 ± 0.06 log CFU/g. The proportion of bacterial spores was an important factor affecting the viability rate of the products during spray drying and storage. The initial spore rate in *P. polymyxa* ZF129 culture was 62%. After spray-drying treatment, the spore rate increased up to 67.35% and 83.66% because vegetative cells could be easily killed by the spray-drying process while spore cells had more tolerance [31]. Meanwhile, considering that the moisture content would influence the cell viability during storage, the moisture content of the microcapsule powder was also investigated. The result showed that the moisture content ranged between 8.00 ± 0.02% and 6.00 ± 0.05%, with a higher inlet temperature leading to lower moisture in those microcapsules. Low moisture content prevents contamination of the product and provides a longer shelf life [32]. The result also agreed with the previous study, which showed that as the temperature of spray drying increased, the moisture content of spray-dried powders decreased [33].

### 3.4. Particle Size Analysis

To investigate the effect of inlet temperature on the size of the ZF129 microcapsules, the size distribution of the three microcapsule samples was measured by the dynamic light scattering technique. The hydrodynamic size distribution of the *P. polymyxa* ZF129 microcapsules with different inlet temperatures (120 °C, 180 °C, and 200 °C) is graphed in Figure 2. The size distribution peak for the three samples was narrow, and a large fraction of particles accumulated at 2–20 μm (Table 4), indicating that the size of microcapsules prepared by the spray drying method is relatively uniform. The terms dv10, dv50, and dv90 represent the cumulative value of particle size volume distribution, indicating that particles smaller than this particle size value in the sample account for 10%, 50%, and 90% of the total sample volume. The mean diameter of three samples (120 °C, 180 °C, and 200 °C) was 8.34 μm, 8.89 μm, and 10.3 μm, respectively. Higher inlet temperatures seemed to create larger microcapsules, which was mostly due to the fast evaporation of solvents during the spray drying procedure [34].

### 3.5. Viability of the Encapsulated ZF129 during Storage

The viability of encapsulated ZF129 with different inlet temperatures (120 °C, 180 °C, and 200 °C) was monitored during storage for 30 days at 4 °C and 25 °C (Figure 3). To make a more effective comparison between the changing trends of ZF129 viability during storage, the viable cell count measured at different storage periods was normalized with the value of the fresh ZF129 powder cell count (Figure 3). After 30 days, the overall decrease in the viability of ZF129 was around 1 logCFU/g, whereas for ZF129 powder, a decrease of up to 2.0–3.0 logCFU/g was observed. The rapid loss of ZF129 viability occurred during the first 2 weeks of storage, and then the curves nearly leveled off until the end of the storage test (Figure 3).

All *P. polymyxa* ZF129 microcapsule powders retained viable cell counts close to or higher than 10^5^ CFU/g after 30-day storage (Figure 3). The highest viability (7.04 log CFU/g) was observed with *P. polymyxa* ZF129 at 4 °C, followed by *P. polymyxa* ZF129 at 25 °C with a residual viability of 6.28 log CFU/g prepared at 180 °C (Figure 3B). The viability of the encapsulated ZF129 microcapsule samples stored at 25 °C was significantly (*p* < 0.05) lower than those stored at 4 °C (Figure 3A–C). Higher storage temperatures cause increased cell metabolism and the death of probiotics [35]. However, the encapsulated ZF129 microcapsules stored at 4 °C achieved the highest viability compared with the encapsulated ZF129 microcapsule cells stored at 25 °C. The loss of viability of free and encapsulated cells at 25 °C is probably due to the oxidation of membrane lipids and the denaturation of proteins that lead to the degradation of macromolecules in bacterial cells [36]. Similar results were also reported in the study of bacterial encapsulation. For example, the stability of spray-dried *L. Paracasei* NFBC 338 was highest during storage at 4 °C, with the viability of probiotic cultures decreasing as the storage temperature increased [33].

### 3.6. The Release Behavior of ZF129 from the Composite Capsules

The release behavior of encapsulated ZF129 in the dissolution medium is presented in Figure 4. A fast release phase can be found at the beginning of 1 h due to the burst effect and high solubility of gum arabic. About 51.5% of ZF129 was released at this stage. Then, during the first 1–24 h, the release rate of viable cells from ZF129 microcapsules slowed down, and 89.5% was released by 24 h. After that, the release curve became flat, and 95.3% was released by 72 h.

### 3.7. Loading of ZF129 in Pelleted Seeds

Herein, we prepared pelleted pak choi seed with ZF129 microcapsule-loaded seeds (ZF129m seeds) and non-encapsulated ZF129-loaded seeds (ZF129 seeds), respectively. The cross section of the seed pelleting layer is shown in Figure 5. The thickness of the pelleting layer was about 200 µm, and microcapsules and free ZF129 cells were embedded in the pelleting layer. The coating bacterial contents of each treatment reached 1.85 × 10^4^ CFU/seed for ZF129 seeds, with the maximum coating bacterial amount of ZF129m seeds being 1.30 × 10^4^ CFU/seed.

### 3.8. Viable Cell Count in ZF129 Microcapsule-Loaded Seed during Storage

To investigate the storage stability of ZF129 microcapsule-loaded seed, the viable ZF129 cell counts on the seed pelleting matrix were tracked for up to 60 days at 25 °C (Figure 6). The number of ZF129 cells was 4.11 log CFU/seed and 4.27 log CFU/seed for ZF129 microcapsule-loaded seeds (ZF129m seeds) and non-encapsulated ZF129-loaded seeds (ZF129 seeds), respectively. The cell counts in both ZF129m seeds and ZF129m seeds declined during the storage period of 60 days. However, the ZF129m seed sample clearly showed a slower decline rate than ZF129 seeds. Specifically, the viable cell of the ZF129m seed samples decreased from 4.11 log CFU/seed down to 3.38 log CFU/seed after 60 days of storage, while the viable cell in ZF129m seeds decreased from 4.27 log CFU/seed down to 1.69 log CFU/seed. The result indicated that the encapsulation of ZF129 with gum arabic can significantly increase cell survival within the pelleting layer, and this can be attributed to the excellent biocompatibility of the gum arabic material. A high survival rate was obtained in ZF129m seeds, which may provide an appropriate environment for the encapsulated cells during storage, possibly resulting in better survival compared with unformulations. Moreover, the microcapsules delayed the survival time and release of cells in a controlled manner, keeping the bacterial content of the ZF129m seed pelletings adequately high for enhanced utilization and extending the utilization period. Similar phenomena were reported in the encapsulation of other bacteria, such as *Lactobacillus acidophilus* [37].

### 3.9. Control Effect of Pak Choi Clubroot Diseases by ZF129m Seed Pelleting in the Greenhouse

To investigate the effects of ZF129-treated seeds against clubroot diseases, ZF129m seeds, ZF129 seeds, and bare seeds were planted in the *P. brassicae*-incubated soil under greenhouse conditions. In the meantime, a group of bare seeds were planted in healthy soil as a healthy control. After growing for 60 days, the phenotype of pak choi root galls with different treatments was photographed (Figure 7). For the infected control group, 83.5% of the pak choi plants were infected with *P. brassicae*, resulting in the formation of large root galls. Both ZF129 seeds and ZF129m seeds effectively suppressed clubroot by alleviating root gall formation, with disease incidences of 56.83 ± 1.22% and 53.2 ± 2.33, respectively (Table 5). The proportions of “scale 3” plants in the group of ZF129 seeds and ZF129m seeds were significantly lower than those in the infected control group. These results were in agreement with the previous study, in which strain ZF129 exhibited significant broad inhibitory spectra against various plant-pathogenic fungi and bacteria and possessed excellent biocontrol characteristics and potential for the biocontrol of vegetable diseases. Besides, the genes responsible for antimicrobial secondary metabolite (e.g., fusaricidin, paenilarvins) synthesis and systemic resistance inducer production were discovered in strain ZF129 [38]. Notably, the control efficacy of ZF129m seeds was significantly higher (71.23%) than the ZF129 group (61.64%). This may be attributed to the protective effect of microcapsules on the ZF129 cells on the surface of seeds. It was well documented that inoculation of the cells of the seed coating agent into the environment may result in poor survival and colonization because of their susceptibility to various environmental factors. Encapsulation of bacteria can effectively protect sensitive bacteria from unfavorable environments. Moreover, the slow-release characteristics of ZF129m seeds enable them to have a longer effective duration than ZF129 seeds.

The efficacy of seed pelleting on pak choi plant growth was also investigated. Phenotype analysis revealed that ZF129m seed produced 2.35% more shoot biomass and 0.6-fold lower root biomass than bare seed (Figure 8). It can be explained that the formation of the clubroot in the control group significantly inhibited the shoot growth of pak choi plants but promoted the accumulation of root biomass. Typically, the shoot/root biomass ratio of the ZF129m seed group is 6.02-fold higher than that of the infected control group, 6.88-fold higher than that of the ZF129 seed group, and 0.78-fold lower than that of the healthy control group. However, it should be noted that the root growth of bare seeds and ZF29 seeds was abnormal, with root lengths of 4.34 cm and 5.21 cm, respectively. Regarding the leaf area, both the ZF129m seed group and the ZF129 seed group were larger than the infected control group. Typically, the mean leaf area of the ZF129m seed group (36.93 cm^2^) was 1.58-fold larger than that of the infected control group (23.44 cm^2^), possibly because the formation of clubroot would prevent the transportation of nutrients from roots to leaves.

## 4. Conclusions

In this work, pak choi clubroot disease was successfully controlled through seed pelleting with encapsulated *P. polymyxa* ZF129. The *P. polymyxa* ZF129 microcapsule powder was prepared by an optimized spray drying method with gum arabic as wall material. The optimized seed pelleting formulas showed good performance in terms of high germination rates and uniform morphology. Pak choi seed pelleted with encapsulated *P. polymyxa* ZF129 (ZF129m seeds) showed better storage stability than that of non-encapsulated *P. polymyxa* ZF129 pelleted seeds (ZF129 seeds). Importantly, the ZF129m seeds also have better bioactivity against clubroot disease, with a control efficacy of 71.23%. Overall, this work not only provided a “seed pelleting with microbial-loaded microcapsules” strategy to control the clubroot disease but also supported the development of biological seed treatments.

## Figures and Tables

**Figure 1 plants-12-03702-f001:**
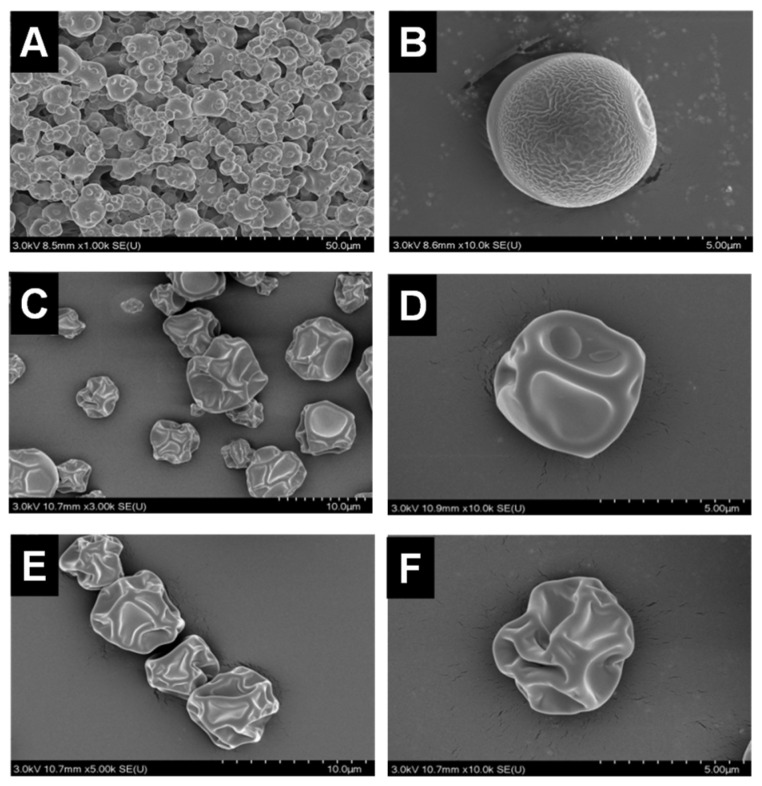
Scanning electron micrographs of the ZF129 microcapsule at different inlet air temperatures (**A**,**B**), 120 °C, (**C**,**D**), 180 °C, and (**E**,**F**) 200 °C at different magnifications.

**Figure 2 plants-12-03702-f002:**
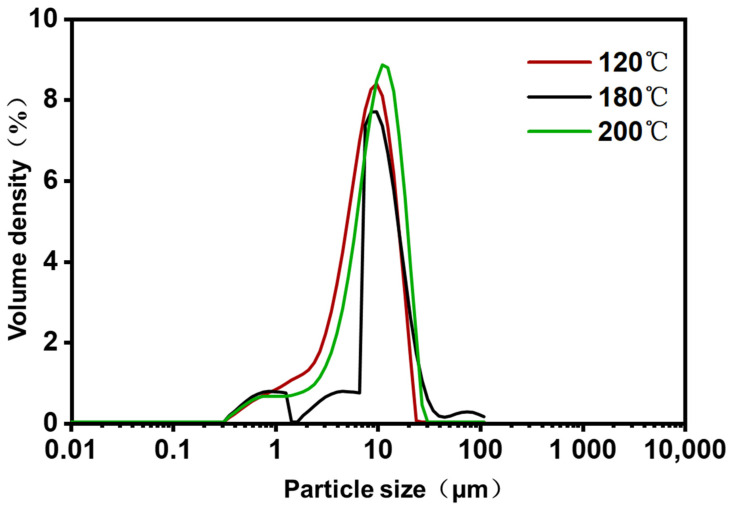
Particle size distribution of *P. polymyxa* ZF129 microcapsules prepared under different inlet temperatures.

**Figure 3 plants-12-03702-f003:**
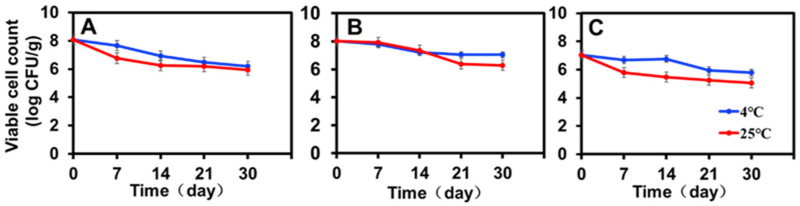
Storage stability of ZF129 microcapsule powders at 4 °C and 25 °C for 30 days. (**A**) *P. polymyxa* ZF129 prepared at 120 °C; (**B**) *P. polymyxa* ZF129 prepared at 180 °C; (**C**) *P. polymyxa* ZF129 prepared at 200 °C.

**Figure 4 plants-12-03702-f004:**
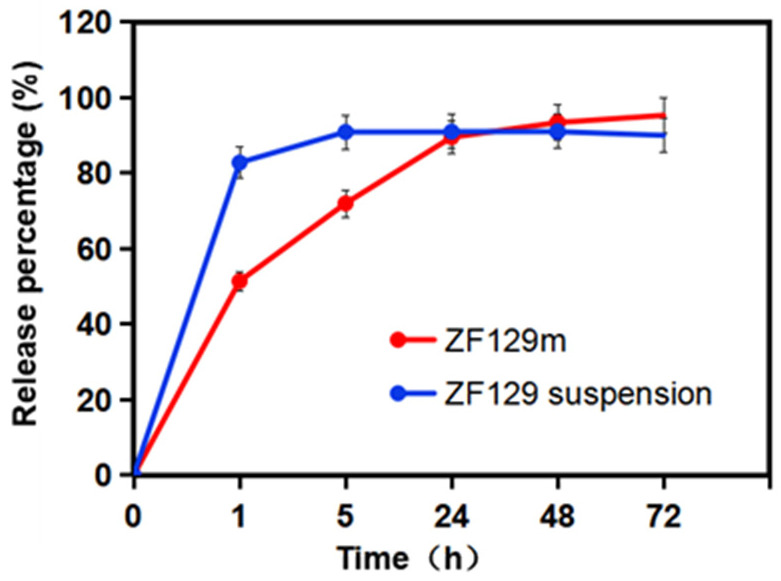
Release profiles of ZF129 microcapsules and ZF129 suspension.

**Figure 5 plants-12-03702-f005:**
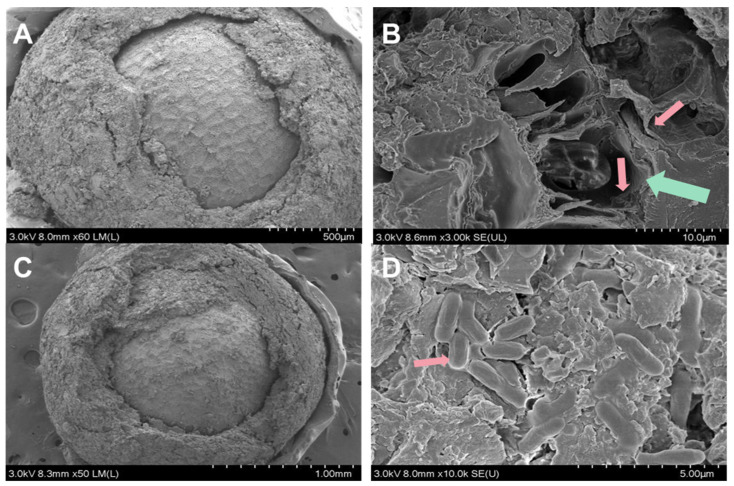
Micrographs of the cross-section of ZF129m seeds (**A**,**B**) and ZF129 seeds (**C**,**D**). The green arrow showed the cross-section structure of the ZF129 microcapsule, and the red arrows showed ZF129 bacteria cells.

**Figure 6 plants-12-03702-f006:**
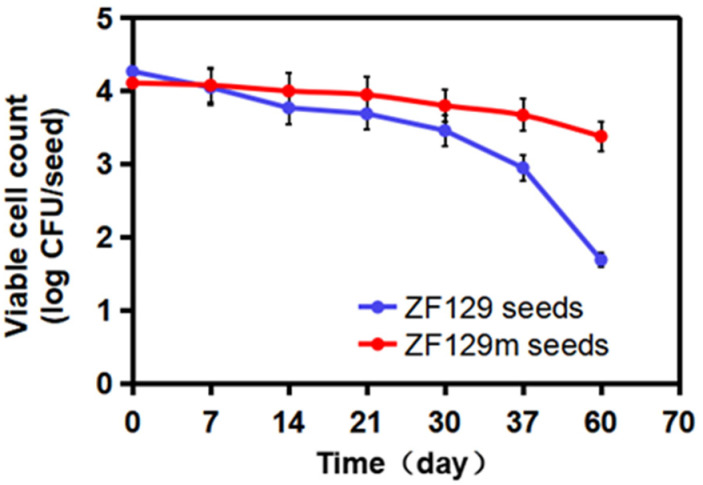
Viability of ZF129 cells on ZF129m and ZF129 seeds after 60 days of storage at 25 °C. Dots represent mean values, and error bars represent the standard deviation of triplicates.

**Figure 7 plants-12-03702-f007:**
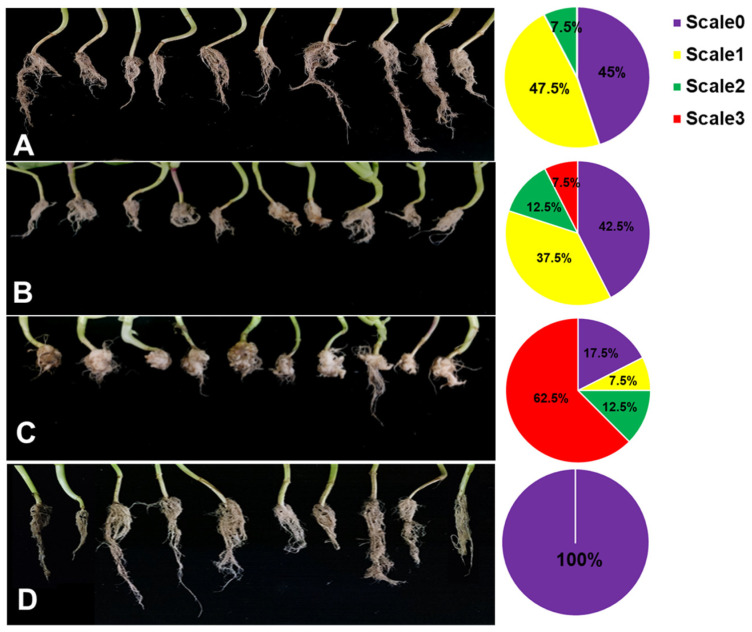
Root phenotype of Pak Choi plants with different treatments. (**A**) ZF129m seeds; (**B**) ZF129 seeds; (**C**) Infected control (non-treated seeds in infected soil). (**D**) Healthy control group (non-treated seeds in healthy soil). The pak choi plants were cultured in *P. brassicae*-infected soil for 60 days. Each treatment consisted of three biological replicates using at least 40 plants for each trial. The pie chart indicates the distribution of the different severity categories for each treatment (different colors indicate different severity scale categories).

**Figure 8 plants-12-03702-f008:**
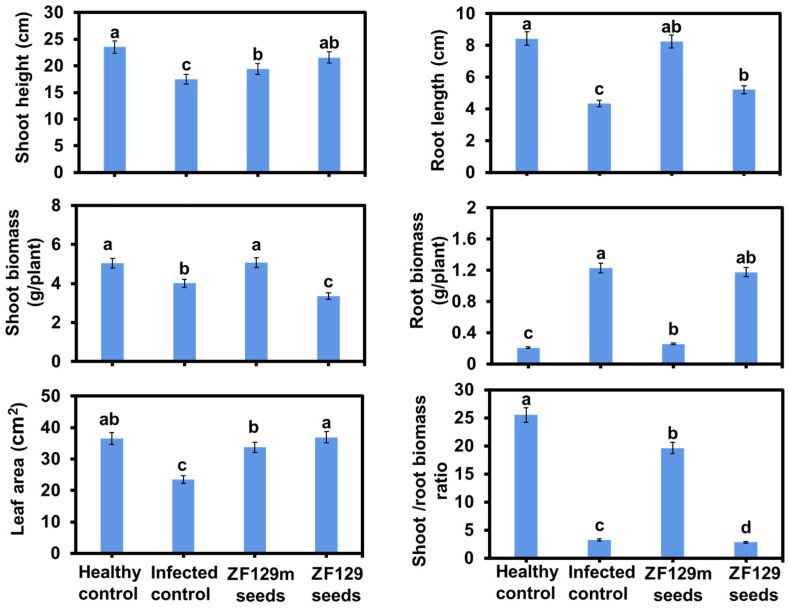
Growth response of pak choi plants with different treatments. All of the plants were cultured with *P. brassicae*-infected soil for 60 days. Each treatment consisted of three biological replicates using at least 40 plants for each trial. The letters above the bars indicate the significant differences (*p* < 0.05), and the error bar represents the SD of triplicates.

**Table 1 plants-12-03702-t001:** Composition of filler formula for seed pelleting (100 g).

Formula	Peat (g)	Talcum (g)	Bentonite (g)	Diatomite (g)
1	50	0	25	25
2	24	0	0	76
3	0	50	0	50
4	23	25	32	20
5	0	0	100	0
6	21	0	35	44
7	50	25	22	3
8	0	25	75	0
9	26	14	60	0
10	0	1	23	76
11	43	10	0	47

**Table 2 plants-12-03702-t002:** Physical property and germination rate of pelleted pak choi seeds with different filler formulas.

Formula	Single Seed Rate (%)	Abscission Rate (%)	Disintegration Rate (%)	Germination Rate (%)
1	98.00 ± 0.50 b	5.00 ± 0.35 f	88.00 ± 0.15 d	76.40 ± 2.10 c
2	98.00 ± 0.30 b	7.00 ± 0.25 f	93.00 ± 0.50 bc	56.80 ± 3.50 d
3	80.00 ± 0.25 c	13.00 ± 0.45 e	74.00 ± 0.25 f	72.70 ± 1.50 c
4	60.00 ± 0.70 d	50.00 ± 0.16 c	92.00 ± 0.55 bc	56.30 ± 2.60 d
5	63.00 ± 0.89 d	53.00 ± 0.18 c	89.00 ± 0.34 c	4.80 ± 0.50 j
6	33.00 ± 0.56 f	66.00 ± 0.34 b	100 ± 0.00 a	32.90 ± 4.20 f
7	98.00 ± 053 b	6.00 ± 0.26 f	44.00 ± 0.54 e	56.80 ± 3.10 d
8	100 ± 0.00 a	24.00 ± 0.56 d	100 ± 0.00 a	92.20 ± 1.10 b
9	100 ± 0.00 a	5.00 ± 0.67 f	100 ± 0.00 a	96.90 ± 2.90 ab
0	57.00 ± 0.51 e	71.00 ± 0.65 a	97.00 ± 0.61 b	56.70 ± 7.20 d
11	94.00 ± 3.50 bc	11.00 ± 0.39 e	82.00 ± 0.57 d	68.60 ± 5.30 cd
Naked seed				98.00 ± 2.30 a

Note: Each value is the mean of three replicates, followed by a standard error. Different letters within each column indicate statistical differences between treatments at *p* < 0.05 (the LSD test).

**Table 3 plants-12-03702-t003:** The effect of different air inlet temperatures (120 °C, 180 °C, and 200 °C) on the encapsulation efficiency, spore rate, and moisture content of *P. polymyxa* ZF129 microcapsule powder.

Inlet Temperature	Encapsulation Efficiency (%)	Bacteria Count (log CFU/g)	Spore Rate of *P. polymyxa* ZF129 Culture (%)	Spore Rate of *P. polymyxa* ZF129 Microcapsules (%)	Moisture Content (%)
120 °C	97.00 ± 0.58 a	8.09 ± 0.05 a	62.07	67.35 ± 0.48 c	8.00 ± 0.02 a
180 °C	96.86 ± 0.66 b	8.03 ± 0.08 a	62.07	82.78 ± 0.59 b	7.60 ± 0.04 b
200 °C	96.17 ± 0.49 b	7.04 ± 0.06 b	62.07	83.66 ± 0.32 a	6.00 ± 0.05 c

Note: Each value is the mean of three replicates, followed by a standard error. Different letters within each column indicate statistical differences between treatments at *p* < 0.05 (the LSD test).

**Table 4 plants-12-03702-t004:** The particle size measurement of the *P. polymyxa* ZF129 microcapsules.

Inlet Temperature	dv10 (µm)	dv50 (µm)	dv90 (µm)
120 °C	2.24 ± 0.48 c	8.34 ± 0.56 c	16.24 ± 0.34 b
180 °C	2.67 ± 0.11 b	8.89 ± 0.47 b	19.41 ± 0.22 ab
200 °C	2.94 ± 0.09 a	10.32 ± 0.36 a	19.61 ± 0.33 a

The data was from measurements using the Mastersizer 3000, and each formulation was produced in one batch of microcapsules. Different letters in the same column indicate a significant difference (*p* < 0.05).

**Table 5 plants-12-03702-t005:** The control efficacy of different seed treatments against clubroot disease in Pak Choi.

Treatment	Disease Incidence (%)	Disease Severity Index	Control Efficacy (%)
ZF129m seed	53.2 ± 2.33 c	21.00 ± 2.19 c	71.23 ± 5.31 a
ZF129 seed	56.83 ± 1.22 b	28.00 ± 2.51 b	61.64 ± 3.45 b
Infected control	83.50 ± 3.12 a	73.00 ± 3.09 a	-

The data represents the mean ± standard deviation. The different letters denoted within the column are statistically different according to the Duncan-multiple range test (*p* < 0.05).

## Data Availability

All data are included in the main text.

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
