# Peer review of "Seed Pelleting with Gum Arabic-Encapsulated Biocontrol Bacteria for Effective Control of Clubroot Disease in Pak Choi"

_plants, 2023, doi:10.3390/plants12213702_

Round 1
Reviewer 1 Report
Comments and Suggestions for Authors
In this study, a method for microbial seed pelleting was developed to protect pak choi seedlings against clubroot disease. Seed pelleting with Paenibacillus polymyxa ZF129 proved to be a effective against clubroot disease in pak choi.The paper is organized well and is interesting for readers.
I have several specific comments on this MS. It should be considered before the paper can be published.
1 I would like see the spore rate in P. polymyxa ZF129 culture and microcapsules. It is important to increase the spore rate for the product quality in storage stability.
2 In the section “ results and discussion”, “Control effect of pak choi clubroot diseases by ZF129m seed pelleting in greenhouse”, additional discussion for the results is necessary, including previous research results on the control effect of clubroot diseases and mechanism.
3 lines 359-360 “Typically, the shoot/root length ratio of ZF129m seed group is 6-fold higher than control group and 6-fold higher than ZF129 seed group”. I did not see the result about “the shoot/root length ratio” in the Figure.
Reviewer 2 Report
Comments and Suggestions for Authors
The use of biocontrol agents to deal with clubroot disease is a highly attractive approach. Here the authors have comprehensively investigated the seed encapsulation procedure for Paenibacillus polymyxa to protect pak choi against infection by Plasmodiophora brassicae. The various parameters and methods for encapsulation are clearly presented alongside some very striking electron micrography of the encapsulated seeds. The protective effect of the selected method is evaluated for resistance to clubroot disease showing that encapsulated seed perform better than un-encapsulated Paenibacillus polymyxa. The paper is clearly written and presented.
Some minor considerations:
What pathotype of P. brassicae has been used?
The method for calculating DSI is mentioned in the methods, could you please also include how control efficacy is being calculated.
In presenting the DSI values in Table 5 it might also be valuable to include pie charts showing the distribution of the different severity categories.
The impact of disease on plant morphology in figure 8 is interesting but without the values from uninfected controls (plants grown in soil not containing P. brassicae spores) it is difficult to appreciate the scale of the protection provide by the encapsulation process.
Comments on the Quality of English Language
There are some minor typos and grammatical concerns, some points that I noticed:
Line 9 “of”or “on” but not both
Line 10 “agents”
Line 12 “to the harsh”
Line 38 “microorganisms have the potential”
Line 105 “pellet” not “pelletize”
Line241 “samples”
Line 250 “higher inlet temperatures seemed to create larger”
Line 251 “evaporation”
Line 257 “temperatures”
Line 277 “Higher storage temperatures cause increased cell metabolism and the death of probiotics”
Line 278 perhaps change “than” to “compared with”
Line 281 “Similar results were”
Line 319 “days” no apostrophe
Line 329 “phenomena”
Line 376 “At the mean time”?
Round 2
Reviewer 1 Report
Comments and Suggestions for Authors
The paper can be published.